# Colorectal Cancer in Inflammatory Bowel Diseases: Epidemiology and Prevention: A Review

**DOI:** 10.3390/cancers14174254

**Published:** 2022-08-31

**Authors:** Elisa Marabotto, Stefano Kayali, Silvia Buccilli, Francesca Levo, Giorgia Bodini, Edoardo G. Giannini, Vincenzo Savarino, Edoardo Vincenzo Savarino

**Affiliations:** 1Gastroenterology Unit, Department of Internal Medicine, IRCCS Ospedale Policlinico San Martino, University of Genoa, 16132 Genoa, Italy; 2Department of Surgery, Oncology and Gastroenterology, University of Padua, 35137 Padua, Italy; 3Gastroenterology Unit, Azienda Ospedale Università di Padova, 35128 Padua, Italy

**Keywords:** colorectal cancer, prevention strategies, prophylaxis, screening, inflammatory bowel disease

## Abstract

**Simple Summary:**

Colorectal cancer (CRC) is one of the most serious potential complications of inflammatory bowel diseases (IBDs). The aging of patients affected by IBDs makes this issue a challenge that will increasingly be faced by clinicians in clinical practice, especially in light of the poorer prognosis for CRC in this group of people when compared with the general population. In this review, we summarize the current epidemiology, risk factors and various prevention strategies proposed for CRC in patients with IBDs.

**Abstract:**

Colorectal cancer (CRC) is currently the third most frequent form of malignancy and the second in terms of mortality. Inflammatory bowel diseases (IBDs) are recognized risk factors for this type of cancer. Despite a worldwide increase in the incidence of CRC, the risk of CRC-related death in IBD patients has declined over time, probably because of successful surveillance strategies, the use of more effective drugs in the management of remission and improved indications to colectomy. This notwithstanding, CRC 5-year survival in patients with IBD is poorer than in the general population. This review provides a summary of the epidemiological features, risk factors and various prevention strategies proposed for CRC in IBD patients. Moreover, there is a special focus on reporting and highlighting the various prevention strategies proposed by the most important international scientific societies, both in terms of chemoprevention and endoscopic surveillance. Indeed, in conducting the analysis, we have given attention to the current primary, secondary and tertiary prevention guidelines, attempting to emphasize unresolved research and clinical problems related to this topic in order to improve diagnostic strategies and management.

## 1. Introduction

Inflammatory bowel diseases (IBDs) are a group of idiopathic conditions, including both ulcerative colitis (UC) and Crohn’s disease (CD), caused by interplays between environmental and patient-related factors which result in dysregulated immune responses directed mainly towards the small and large bowel [1]. While in UC the damage is limited to the colonic mucosa, in CD it is transmural and can involve any segment of the digestive system from the mouth to the perianal area [2].

The persistent relapse and remission levels of inflammation in IBDs are responsible for most of their complications, including, above all, colorectal cancer (CRC)—one of the most feared complications since its first description [3,4]. Indeed, IBD-associated CRC arises from a specific carcinogenic pathway involving chronic inflammation which is distinct from the traditional adenoma–carcinoma and serrated adenoma pathways [5].

Globally, CRC is currently the third most frequent form of malignancy and the second in terms of mortality, and its incidence has steadily increased worldwide over the past 40 years [6]. It places a serious burden on both patients and national health systems, since it represents the second leading cause of cancer-related disability-adjusted life-years (DALYs) and is responsible for high direct and indirect costs in developed and developing countries [7].

In patients affected by UC and CD, the risk of CRC-related death seems to have declined over time [8]. Successful surveillance strategies, more effective therapies and better indications to colectomy are likely responsible for this trend [9]. This notwithstanding, patients with IBDs affected by CRC have poorer 5-year survival than the general population [6]. Strong evidence-based prevention strategies are needed in order to mitigate the burden of this malignancy among these patients [10].

In the last few years, several innovations that have modified the management of patients with IBDs have been the subject of experimentation. These innovations are mainly represented by advancements in therapeutics and endoscopic techniques, although their roles have not yet been fully included in international guidelines. In this review, we summarize the current epidemiology, risk factors and potential prevention strategies for IBD-related CRC, a distinct condition from sporadic CRC, for information on which we refer to the numerous reviews already available in the literature. Our aim is to deliver to the reader an updated key for interpretation, after an in-depth analysis of the ample and heterogeneous literature on this topic.

## 2. Literature Review

In order to find relevant studies, a computerized (PubMed, Embase^®^ and Medline) and manual literature search was carried out, which ended in May 2022, with particular focus on the past 12 years, using search terms referring to IBDs, onset of CRC in patients with IBDs and IBD-related CRC epidemiology and prevention. The detailed web research can be found in the Appendix A. Non-original research, such as editorials and commentaries, were excluded. Databases were last accessed on 30 May 2022. The full texts of the included studies were obtained with institutional access or open-access licenses and further reviewed to screen for the most relevant manuscripts.

## 3. Epidemiology

### 3.1. Epidemiology of Sporadic CRC

The incidence of CRC has been increasing over time. According to the 2020 Global Cancer statistics, CRC is currently the third most frequent form of malignancy in both males and females, with 1.9 million new diagnoses, while it ranks second in terms of mortality, with 935,000 deaths [11]. CRC incidence is four times higher in developed than in emerging countries, while mortality rates seem to be comparable due to a worse case–fatality rate in developing economies. CRC is more frequent in Europe, Oceania and North America, while the incidence tends to be low in most regions of Africa and Asia [11]. The occurrence of CRC is responsible for a significant decrease in both life expectancy and quality of life. Indeed, it caused 24 million disability-adjusted life-years (DALYs) globally in 2019, with an age-standardized rate of 295.5 (275–316) DALYs per 100,000 person-years, showing a clear declining rate between 1990 and 2019 [6].

### 3.2. Epidemiology of IBDs

The incidence and prevalence of IBDs have been increasing in recent decades, worldwide [12]. Currently, over one million subjects in the US and 2.5 million in Europe are affected by IBDs, with an estimated prevalence of 0.5% in the general population in the Western world [13]. The highest prevalence of IBDs is reported in Europe, with 505 cases of UC per 100,000 inhabitants in Norway and 322 cases per 100,000 of CD in Germany, followed by 286 cases of UC per 100,000 in the USA and 319 cases of CD per 100,000 in Canada. Despite the lower availability of epidemiological data in newly industrialized countries, recent studies have shown an increased prevalence of IBD in South America, Eastern Europe, Asia and Africa [14].

### 3.3. Epidemiology of CRC in Patients with IBDs

It is now widely demonstrated that patients with longstanding colonic IBD have a higher risk of developing CRC when compared with the general population [15,16,17,18]. Its incidence varies according to geographical distribution, with higher rates in the US and UK, and lower incidence in Scandinavian countries [17]. The worldwide incidence rate of CRC in CD is estimated to be between 19.5 and 344.9/100,000 per year, and between 54.5 and 543.5/100,000 per year in patients with UC [19]. The standardized incidence ratios (SIRs) for developing CRC in Europe and in the US are, respectively, 1.9 and 3.4 times higher in those with CD than in the general population, while these figures are 2.4 and 5.2 times higher in patients with UC [4,20]. In Asia, although data are limited, the prevalence of IBD-associated CRC is lower than in other regions [21]. The roles of ethnic origins and geographical location have still to be investigated, but it has to be emphasized that the abovementioned estimates should be considered in the context of differences in the availability of IBD therapies, surveillance practices, access to specialized gastroenterological care and diets. For instance, lower access to more efficacious medical treatments could determine a higher rate of patients who undergo surgery, thus decreasing the likelihood of cancer.

Furthermore, CRC-related mortality is higher in those affected by IBDs. In particular, when compared with the general population, the rate is 1.4 times higher both in CD and UC patients [22,23]. Data on indeterminate colitis (IC) are lacking: only one study has shown that the risk of developing CRC in patients with IC is higher than in patients with UC [24].

As far as time trends are concerned, the incidence of CRC in patients with UC has been declining over the last few decades, from 4.29 per 1000 per year to 1.21 per 1000 per year [25], while in patients with CD it seems to have been stable over time [26].

CRC plays a role in the natural history of these inflammatory diseases, since it represents one of the main causes of death in these patients (15% of all deaths in IBD patients), while IBD-associated CRC represents only 1–2% of all cases of CRC in the general population [27].

Comparing sporadic and IBD-associated CRC, there appear to be several differences in terms of age at diagnosis, the segment of the colon involved and mortality rates. Diagnosis of CRC in those with IBDs occurs earlier in life, with a mean age at diagnosis of 50–60 years compared to 65–75 years in sporadic CRC [28]. Localization of the underlying IBD affects directly the risk of CRC development. In those with colonic CD, it is four times higher than in those with a pure ileal involvement [29]. In particular, the right colon is the segment more frequently affected by CRC in these patients [30]. Despite being previously debated in the literature, recent studies have demonstrated that patients affected by IBD-associated CRC have a 1.2 to 2 times higher risk of death and a shorter overall survival than those with sporadic cancer [31,32,33]. A Japanese study focused on differential survival on the basis of cancer stage and showed that those with UC and stage III CRC had worse prognoses than patients with sporadic CRC, while no survival differences were observed in patients with earlier cancer stages [34]. Prognosis is influenced, also, by age. In patients with IBDs aged above 65 years, 5-year survival is similar to those with sporadic CRC, while in those aged below 50 years it is appreciably lower in patients with IBDs than in the general population (58.8% vs. 71.4%, *p* < 0.001) [35].

Despite these findings, it seems that the risk of death from CRC in patients with IBDs has been decreasing over time [4,36].

## 4. Risk Factors

In order to improve patient prognosis and quality of life, the optimization of primary prevention strategies is crucial. The identification of risk factors involved in the development of CRC in patients with IBD represents an essential step in this process.

Risk factors can be categorized into “patient-related” factors, such as young age at diagnosis of IBD (<20 years), male gender and family history of CRC (especially in patients aged <50 years), and “disease-related” factors, such as extension of colitis and its duration (>10 years), concomitant history of primary sclerosing cholangitis (PSC) and presence of endoscopic and histologic inflammation (including post-inflammatory polyps) [37,38,39,40].

In particular, duration of disease greatly influences the risk of CRC both in patients with UC (cumulative risk of 2%, 8% and 18% after 10, 20 and 30 years of disease duration) and in those with CD (cumulative risk of 2.9% at 10 years from diagnosis) [17,29]. Among all patients with IBDs, the cumulative risk of developing CRC reaches 1%, 2% and 5% after 10, 20 and more than 20 years of disease duration, respectively [41]. In a recent review of the literature on the development of CRC in patients with UC with low-grade dysplasia (LGD), the annual incidence of progression to CRC was reported as being 0.8%, and the risk of CRC was higher when LGD was confirmed by an experienced pathologist (i.e., 1.5%) [42].

Furthermore, the inflammatory activity of underlying IBD plays a central part in CRC development, generating oxidative-stress-induced damage to DNA that may activate tumor-promoting and disable tumor-suppressing genes [30].

## 5. Primary Prevention

Many therapeutic approaches are used in the treatment of UC and CD. Some of these drugs could prevent the development of CRC in patients with IBD, not only reducing the activity of inflammation but also targeting mechanisms involved in carcinogenesis. The chemopreventive effects of the principal drugs used in IBDs are summarized in Table 1.

### 5.1. 5-Aminosalicylic Acid Compounds

Besides its well-known anti-inflammatory effects, 5-Aminosalicylic Acid (5-ASA) may prevent CRC by interacting with specific molecular mechanisms, for instance, by inhibiting activation of the transcription of nuclear factor kB (NF-kB) [65], downregulating cyclooxygenase-2 (COX-2) and inhibiting phospholipase D activity and proliferation [66,67]. Four meta-analyses have shown significant reductions in the occurrence of CRC in patients with UC but not in those with CD [43,48]. 5-ASA showed protective effects especially with doses of >1.2 g/day [44,45]. It is important to point out that its protective effect is still characterized by a low level of evidence in the literature [68]. Indeed, just a single meta-analysis reported a CRC-protective effect of 5-ASA for both clinical (relative risk (RR): 0.46, 95% CI 0.34–0.61) and population-based (RR: 0.70, 95% CI 0.52–0.94) studies [46]. A conflicting review did not show a significant protective effect against CRC occurrence in patients with IBD [69].

The current European and British guidelines recommend the use of mesalamine compounds in patients with UC for the chemoprevention of CRC, while the American guidelines do not emphasize this, underlining that only appropriate secondary prevention is crucial [16,47,70].

### 5.2. Thiopurines

Thiopurines, consisting of azathioprine (AZA) and mercaptopurine (MP), are used to treat patients with IBDs in order to maintain long-term glucocorticoid-free responses [71,72].

Their specific mechanism of action in preventing CRC is still unknown, but several studies indicate that they could act either by blocking the TcR signaling pathway, which leads to T-cell apoptosis, or by inhibiting purine synthesis [73]. Their protective effect appears to be greater in patients with longstanding (>8 years) and extensive colitis [49,52,74]. On the other hand, a meta-analysis by Jess et al. failed to demonstrate a chemopreventive effect in either UC or CD [50]. In line with that, a 2017 case–control study nested in the CESAME cohort demonstrated a significant decrease in the risk of CRC in patients with IBDs who received aminosalicylates but not in those who were exposed to thiopurines [51]. On the basis of the above-mentioned studies, data supporting the role of thiopurines in the chemoprevention of CRC in patients with IBD seem to be less consistent than those published on 5-ASA.

The current European guidelines neither recommend nor disrecommend chemoprevention with thiopurines, while the British and American guidelines suggest that they may have a role in UC treatment [16,47,70]. No established conclusions can be drawn concerning the chemopreventive role of thiopurines in CD because of the limited data available [75].

### 5.3. Anti-TNFα Agents

Anti-TNF-α agents are routinely used to induce and maintain remission in patients with moderate-to-severe UC and CD [71,72]. They act by activating the NF-kB transcription factor family, which leads to an innate immune response and apoptotic response of leukocytes in the lamina propria [76]. By improving longstanding chronic inflammation, they could reduce colonic dysplasia and carcinogenesis. Popivanova et al. demonstrated that blocking TNFα in animal models could reduce carcinogenesis associated with chronic colitis [55]. A recent study by Alkhayyat et al. showed a reduced rate of CRC in patients treated with anti-TNFα, with and without immunomodulators [53].

Furthermore, two nation-wide studies found a significant protective role for anti-TNFα in this setting. On the other hand, similarly to thiopurines, other studies found no association between anti-TNFα exposure and risk of developing CRC among patients with IBDs [54,77].

Further prospective long-term studies are needed to support these data; therefore, international guidelines do not recommend anti-TNFα drugs as chemopreventive agents. Moreover, there are only scarce data about the chemoprotective effects of newer biologic agents, such as Vedolizumab and Ustekinumab.

### 5.4. Ursodeoxycholic Acid (UDCA)

Available data on the possible role of UDCA in chemoprevention of CRC in patients with IBD are controversial. UDCA could reduce the risk of developing CRC in patients with UC and PSC by decreasing colonic concentrations of secondary bile acids that may act as carcinogens [58]. Notably, a study by Eaton et al. showed that the risk of developing CRC was higher in patients who received high doses of UDCA (28–30 mg/Kg) than in those who were exposed to placebo (HR 4.44, 95% CI 1.30–20.10; *p* = 0.02) [59].

### 5.5. Dietary Compounds and Lifestyle Habits

There are well-established risk factors for sporadic CRC, such as smoking alcohol use and red meat consumption, which can cause CRC in different ways [78]. Red meat is a source of iron porphyrin pigment, which is responsible for the induction of carcinogenesis through the formation of nitroso compounds. Furthermore, it can cause the activation of insulin and insulin growth factor-1 receptors and may lead to increased cell proliferation and reduced apoptosis [79]. In addition, heterocyclic amines produced by cooking red meat at high temperatures can contribute to carcinogenesis [80].

There is no specific diet proven to have a chemopreventive role in IBD patients, though foods rich in anthocyanins (such as strawberries and black raspberries) have been shown to have potential chemopreventive effects [81].

### 5.6. Statins

Statins are routinely used in the treatment of hypercholesterolemia, and they act by inhibiting 3-hydroxy3-methylglutaryl-coenzyme A (HMG-CoA), which is involved in endogenous cholesterol biosynthesis. In addition, they seem to have some other pleiotropic effects, such as reducing inflammation by interacting with integrin LFA1, inducing apoptosis and modulating angiogenesis [82].

Their potential chemopreventive effect on the occurrence of sporadic CRC has been widely evaluated in several previous studies [83]. However, conflicting results have been reported for IBDs. In particular, inverse associations between the occurrence of IBD-associated CRC and the use of statins have been found in a large cohort study including 11,001 patients with IBDs and a case–control study by Samadder et al. [60,61]. However, the latter had some major limitations, such as self-reported IBD diagnosis and a lack of information about the dose or duration of statin therapy [61]. On the other hand, a conflicting, large retrospective study did not find a significant chemopreventive effect for statins in patients with IBDs [62].

### 5.7. Vitamin D

In the last few years, there has been growing interest in the potential anti-inflammatory role of Vitamin D.

1α, 25-dihydroxyvitamin D3 {1,25-(OH) 2D3}, the active form of vitamin D, plays a crucial role in maintaining mineral homeostasis. Besides its well-known effects on bones, it exerts anti-inflammatory and growth-inhibitory actions [84].

Vitamin D intake and sporadic CRC seem to be inversely associated [63]. Several studies also suggest a link between IBD development and Vitamin D receptor polymorphisms [85,86].

Both human and animal studies suggest that vitamin D could prevent and reduce inflammation in this group of patients, thus playing a possible chemopreventive role in IBD-associated CRC [64,87,88].

Given its tolerability, few side effects and low cost, further studies are needed to figure out its true role in chemoprevention in patients with IBD.

### 5.8. Gut Microbiome Composition

There are some bacteria which seem to be frequently associated with sporadic CRC, such as *Escherichia coli*, *Streptococcus gallolyticus*, *Enterotoxigenic Bacteroides fragilis*, *Enterococcus faecalis* and *Fusobacterium nucleatum* [89]. Since the gut microbiome is linked to chronic inflammation in IBDs, it could have a role in sustaining carcinogenesis in these patients [90].

The alterations in the gut microbiome involved in IBD pathogenesis (increased presence of *Caudovirales*, *Clavispora lusitaniae*, *Pasteurellaceae*, *Veillunellaceae*, *Fusobacterium* species, *Ruminocuccus gnavusa*, *Proteobacteria* and *Escherichia coli*) could allow for different therapies with lower toxicity profiles for patients, such as probiotic and prebiotic agents, which could act as immunomodulators [91].

Gut alterations appear to have a role also in IBD-CRC development [92,93]. In particular, *E. coli* is considered to play a part in the induction of both chronic inflammation in IBD and IBD-associated CRC. Its lipopolysaccharides increase the expression of Toll-like Receptor 4 (TLR4)—a known and recognized step in IBD-CRC tumorigenesis [94]. It is also responsible for NF-kB over-expression, which is a contributor to inflammation and CRC development. An aggressive adherent invasive *Colibactin equipped E. coli* is more prevalent in the colonic mucosa in patients affected by CD and UC [95]. Other bacteria, such as *Streptococcus bovis* and *Fusobacterium nucleatum*, are increased in the tumor microenvironment and could raise inflammatory levels, contributing to the development of IBD-associated CRC [96].

Real-life studies specifically directed at the evaluation of how gut alterations may impact IBD-associated CRC are needed.

## 6. Secondary Prevention

Lower endoscopy is the only method used for secondary prevention of CRC in patients affected by IBDs, since it allows the direct evaluation of colonic mucosa and the possibility of performing biopsies for histological examination. Thus, endoscopic surveillance turns out to be an important weapon with which to achieve an early diagnosis. Historically, since the signs and symptoms of CRC can be confused and overlap with not fully controlled UC or CD, diagnosis can be delayed, leading to more complicated management of the disease and a worse prognosis for patients [97].

Planning a surveillance campaign with repeated colonoscopies in patients with IBD colitis has several purposes, including that of improving the prognosis and survival of patients by the early identification of precancerous lesions or early-stage cancers and avoiding unnecessary prophylactic proctocolectomies in order to protect patients’ quality of life [98].

### 6.1. Open Surveillance Issues

There are several open issues regarding the traditional surveillance method involving repeated colonoscopies. The first is that its effectiveness has never been demonstrated in randomized clinical trials, and therefore its efficacy in reducing the risk of death in patients with IBD and CRC is mainly supported by indirect evidence [99]. There is, however, evidence that CRC in surveilled patients is usually diagnosed at an earlier stage and that patients consequently have a better prognosis, but this can be influenced by a lead time bias [100]. It must be considered that patients with IBD are subjected to colonoscopies more frequently than the general population because of symptoms or assessment of response to treatment, and thus they are more frequently diagnosed with interval cancers. These occur in 16% of IBD patients compared to 6% in the general population [101]. Lastly, surveillance with repeated colonoscopies places not negligible burdens on both patients and gastroenterologists, since the first must be compliant with multiple appointments, while the second must perform long and complex endoscopic examinations with numerous biopsy samples. If these colonoscopies are not performed correctly, the effectiveness of the procedure, and therefore of the entire surveillance program, loses significance.

### 6.2. Timing of Surveillance

In order to define the best surveillance strategy, it is essential to understand the correct timing at which colonoscopies should be performed. The initial timing of endoscopic screening is crucial. Indeed, CRC diagnosis is delayed or missed in 17% to 28% of patients when surveillance is started 10 years from IBD diagnosis, as previously stated in most international guidelines [102,103,104]. In addition, even when starting at 8 years from IBD diagnosis, an important number of CRCs could already be lost [39,105]. This has underlined the importance of the initiation of surveillance, which must take place at 8 years from the moment of the first appearance of symptoms and not from the date of diagnosis, since these two moments can be very distant from each other in patients with IBD [106]. For this reason, most international guidelines now recommend that screening colonoscopies start 8 years after the onset of symptoms [16,107]. Another aspect to keep in mind regarding the beginning of surveillance is the possible presence of concomitant PSC; where this is present, surveillance should be started when PSC is diagnosed [108].

The interval between surveillance colonoscopies should be defined on the basis of several criteria. Firstly, the activity of underlying IBD should be taken into account, as it is advisable to perform endoscopies during periods of remission, since acute inflammation may jeopardize the interpretation of dysplasia at histological evaluation [109]. Following screening endoscopy, the evaluation of the correct timing of the subsequent controls must be guided by a multimodal stratification of the risks [15]. In particular, the presence of strictures, dysplasia within the previous 5 years, concomitant PSC, extensive colitis with severe inflammation and a family history of CRC in first-degree relatives <50 years put the patient in the highest risk profile [41,108,110,111]. The presence of extensive colitis with mild/moderate inflammation, post-inflammatory polyps and family history of CRC in first-degree relatives >50 years represent intermediate risk characteristics, with the remainder of patients falling into the low-risk category [15,16,112]. Based on risk categories, intervals between surveillance colonoscopies vary between 1 and 5 years among European scientific societies [16,113], and between 1 and 3 years among US scientific societies [47,114]. Societal interval recommendations for endoscopic surveillance are summarized in Table 2.

Patients with CD isolated to the small bowel have a risk of CRC that is comparable to that of the general population, and thus traditional CRC surveillance recommendations should be followed [29]. Lastly, patients affected by isolated proctitis should not undergo screening colonoscopies [34].

### 6.3. Optimal Endoscopic Technique

The optimization of the endoscopic technique is crucial in the adequate management of surveillance for CRC in patients with IBD. Firstly, an effective bowel preparation with optimal visualization of the colonic mucosa is essential for a high-quality surveillance colonoscopy, since it significantly affects the lesion detection rate [115,116,117,118]. The biopsy sampling technique recommended until a few years ago consisted in performing four random biopsies every 10 cm and biopsies on any suspicious lesion [114,119]. The indications derived from a mathematical model described in one study demonstrated that 33 or more non-targeted jumbo forceps biopsies could detect dysplasia with 90% confidence [120]. In recent years, we have witnessed a real paradigm shift in relation to this issue with the transition from random to targeted biopsies. In fact, considering that the colorectal surface amounts to about 2700 cm^2^ and the biopsy surface to 0.2 cm^2^, carrying out 40 random biopsies, as suggested by guidelines, accounts for only 0.03% of the large bowel surface [121]. For this reason, several useful methods have been developed to better identify possible suspicious lesions that could indicate the locations at which to perform targeted biopsies. The magnification of images was first achieved through the use of dye-spray chromoendoscopy (DCE). This technique was able to achieve a two-fold higher identification of dysplastic lesions than standard white light endoscopy (WLE), and a recent meta-analysis reported a 1.6-fold higher dysplasia detection rate with DCE than with HD-WLE [122]. The impact of DCE on long-term CRC risk and survival has not been evaluated thus far. In addition, virtual chromoendoscopy (VCE) systems, such as Olympus “NBI imaging”, Pentax “i-scan” and Fuji “Fujinon”, showed analogous results for lesion detection rates, with shorter withdrawal times compared with DCE [123,124,125]. In particular, a randomized controlled trial by Leifeld et al. showed a similar dysplasia detection rate between WLE with 40 random biopsies and NBI with 10 random biopsies, with the latter being associated with fewer specimens (11.9 vs. 38.6, *p* < 0.001) and a shorter withdrawal time (23 vs. 13 min, *p* < 0.001) [126]. In addition, dysplasia is difficult to find in colonic segments that are not inflamed [127].

In light of these results, the European Society of Gastrointestinal Endoscopy (ESGE), the European Crohn’s and Colitis Organisation (ECCO), the American College of Gastroenterology (ACG) and the 2021 SCENIC update now recommend VCE as an alternative to DCE [128,129].

Recently, artificial intelligence (AI) has been used to detect polyps in colonoscopies. No studies have been conducted with the aid of AI in colorectal cancer surveillance in patients with IBD so far [130].

### 6.4. Management of Dysplasia Detection

The histological evaluation of suspected dysplasia should be confirmed by two independent expert pathologists, in light of the great importance such findings have in the clinical management of these patients [15,131]. If confirmed, management depends on grade (LGD vs. HGD), resectability and the endoscopic visibility of the lesion.

Colectomy is necessary in case of unresectable visible dysplasia or HGD or invisible multifocal dysplasia, while endoscopic polypectomy should be chosen if lesions can be resected [9,128]. In the case of detection of dysplasia on an invisible lesion, repeat examination with DCE or VCE should be carried out with extensive nontargeted biopsies in the area of prior dysplasia, and, if confirmed, a new colonoscopy with CE should be performed within three months [15].

## 7. Tertiary Prevention

Patients with IBD who have undergone colectomy for CRC rarely develop a new malignant lesion in the ileal pouch (only 1.3% after 20 years) [132]. Due to this low incidence, the necessity of endoscopic pouch surveillance is debated, and no consensus exists [99]. Nevertheless, the presence of PSC and chronic pouchitis represent risk factors for recurrence; thus, these patients should be considered for annual surveillance of the ileal pouch [111]. No studies exist so far on the use of chromoendoscopy in pouch surveillance.

A role for chemoprevention in tertiary prevention of sporadic CRC has recently been suggested in the literature. In particular, evidence of the association between low-dose 5-ASA and enhanced CRC survival is increasing [133]. Moreover, some authors have started designing randomized clinical trials (RCTs) with the aim of evaluating the potential role of this molecule in tertiary prevention [134]. Similarly, observations of the beneficial role of physical activity in patients already diagnosed with CRC have led to its promotion [135]. Studies dealing with the role of cancer-prevention agents in tertiary prevention in patients affected by IBDs are lacking so far.

## 8. Discussion and Future Perspectives

Recent studies in the literature have shown new trends in the epidemiology and prevention of CRC in patients affected by IBDs. The mortality rate of this malignancy in this group of patients, even if higher than in sporadic CRC, has been declining in the last few decades. Prospective RCTs regarding chemoprevention in those affected by IBDs are lacking and data in the literature are scarce and partial. This notwithstanding, anti-inflammatory agents, such as 5-aminosalicylate compounds and immune modulators, have been considered as potential chemopreventive agents. Endoscopic surveillance strategies should be evaluated carefully, both in terms of timing and endoscopic technique. All of the most influential gastroenterological scientific societies have endorsed CCR surveillance starting 8–10 years after the onset of symptoms. Techniques for image magnification (VCE and DCE) were found to be superior to traditional WLE and are recommended. In Table 3, a summary of key messages about the epidemiology and prevention of CRC in IBDs is presented.

Further research is needed in order to improve the care of aging patients with IBDs. Recently identified oncogenic gut microbiota and molecular biomarkers could become useful auxiliary tools in sporadic and IBD-associated CRC treatment [136,137]. Interesting advancements could also be made in endoscopic techniques with confocal laser endomicroscopy (CLE), which potentially would be able to better detect foci of dysplasia [138].

## 9. Conclusions

Evidence is accumulating regarding the importance of a number of factors capable of influencing the risk of CRC development in patients with IBDs. In order to ensure an optimal CRC prevention strategy, it is essential to adopt a patient-tailored approach. Stratifying risks for both patients and disease characteristics allows the identification of subgroups of patients who need closer surveillance and more intensive treatment.

Finally, shared and evidence-based screening programs for CRC in patients with colonic IBD should be defined, but their cost-effectiveness must be accurately evaluated, in light of the relatively low incidence of IBD-associated CRC. 

## Figures and Tables

**Table 1 cancers-14-04254-t001:** Summary of the principal studies on CRC chemoprevention in IBD patients.

Medication	Study Design	Results
**5-Aminosalicylic Acid** **Compounds**	Systematic reviews with meta-analysis [43,44,45,46,47]	Protective effect against CRC, especially for doses > 1.2 g. One study [44] did not show a protective role but it included heterogeneous studies
	Meta-analysis [48]	Protective effect (OR 0.70; 95% CI 0.54–0.92)
**Thiopurines**	Prospective observational study [49]	Protective effect of thiopurines in long and extensive colitis on occurrence of both HGD and CRC
	Meta-analysis [50]	Not significant protective effect on dysplasia/CRC occurrence reduction (OR 0.87; 95% CI 0.71–1.06). Great heterogeneity across the studies in terms of differences in outcomes (ranging from neoplasia to severe neoplasia or CRC alone)
	Case–control study [51]	Protective effect against CRC occurrence in patients exposed to salicylates (OR 0.59; 95% CI 0.37–0.94) but not in those who received thiopurines (OR 0.76; 95% CI 0.43–1.34)
	Systematic review and meta-analysis [52]	Protective effect (OR 0.49; 95% CI 0.34–0.70). However, great heterogeneity across the studies, specifically in terms of thiopurine exposure
**Anti-TNFα**	Retrospective cohort study [53,54]	First [52] showed a protective effect against both UC and CD. The second [55] found no significant association with CRC
	Population-based cohort study [56]	Significant decrease in CRC in patients with longstanding UC
	Case–control study [57]	Protective against CRC occurrence in patients with IBD
**Ursodeoxycholic Acid** **(UDCA)**	Clinical trial [58]	Protective effect of UDCA against both dysplasia and CRC
	Randomized, double-blind, placebo-controlled trial [59]	UDCA at high doses in patients with PSC, and UC was associated with a higher rate of CRC compared with placebo (HR 4.44; *p* = 0.02)
**Statins**	Meta-analysis [60]	Modest reduction in sporadic rectal but not colon cancer risk (RR = 0.90, 95% CI 0.86–0.95); long-term use (>5 years) does not affect risk (RR = 0.96, 95% CI 0.88–1.04, *p* = 0.297)
	Retrospective population-based study [61]	Inverse association with IBD-associated CRC in Ashkenazi Jewish population
	Retrospective cohort study [62]	No association between occurrence of CRC in patients with IBD and statin exposure
**Vitamin D**	Review [63]	Vitamin D levels and sporadic CRC are inversely associated. Data from animals and cell cultures support its chemopreventive role
	Review [64]	Vitamin D ameliorates chronic inflammation in IBDs and could have a role in preventing carcinogenesis

OR = Odds Ratio; HR = Hazard Ratio; CI = Confidence Interval; CD = Crohn’s sisesase; CRC = colorectal carcinoma; HGD = high-grade dysplasia; IBD = inflammatory bowel disease; PSC = primary sclerosing cholangitis; UC = ulcerative colitis.

**Table 2 cancers-14-04254-t002:** Societal recommendations for endoscopic surveillance.

Society	Low Risk	Intermediate Risk	High Risk
	All other cases	Extensive colitis with mild/moderate inflammation, post-inflammatory polyps, family history of CRC in first-degree relatives >50 years	Stricture, dysplasia within past 5 yr PSC, extensive colitis with severe inflammation, family history of CRC in first-degree relatives <50 years
**ACG 2019** [70]	Every 1–3 yr	Adjust intervals on the basis of previous colonoscopies and combined risk factors	Every year
**AGA 2010** [114]	Every 1–3 yr After two negative exams	Every 1–2 yr	Every year
**BSG 2010** [113]	Every 5 yr	Every 3 yr	Every year
**ECCO 2017** [16]	Every 5 yr	Every 2–3 yr	Every year

CRC = colorectal carcinoma; PSC = primary sclerosing cholangitis; ACG = American College of Gastroenterology; AGA = American Gastroenterological Association; BSG = British Society of Gastroenterology; ECCO = European Crohn’s and Colitis Organisation.

**Table 3 cancers-14-04254-t003:** Epidemiology and prevention of CRC in IBDs; key messages divided per topic.

**Epidemiology**
**Epidemiology of sporadic CRC**: CRC is the third most frequent form of malignancy and the second in terms of mortality and cancer-related DALYs; its incidence is increasing worldwide.
**Epidemiology of IBDs**: IBDs’ incidences and costs have been increasing in the last few decades. Clinicians and national health systems will increasingly have to deal with these conditions.
**Epidemiology of CRC in patients with IBDs**: Risk of IBD-associated CRC is higher in UC than in CD. Thanks to increased adherence to endoscopic surveillance and the improved quality of endoscopy and clinical management, its incidence is now decreasing.
**Risk Factors**
These can be divided into: **Patient-related factors**: young age at diagnosis (<20 years), male gender and family history of CRC; **Disease-related factors**: extension of colitis and its duration (>10 years), concomitant PSC and inflammatory activity.
**Primary prevention**
**5-Aminosalicylic Acid compounds**: 5-aminoacylates can be reasonably regarded as chemoprevention tools in association with proper endoscopic surveillance. Therefore, their long-term use should be encouraged.
**Thiopurines**: Thiopurines’ chemopreventive effects are not supported by strong clinical evidence. Furthermore, non-melanoma skin and lymphopoietic cell cancers are known side effects of their prolonged use.
**Anti-TNFα agents**: There is not sufficient evidence to support clear protective effects. Further findings are needed to analyze their potential chemopreventive role in patients with IBD. Therefore, international guidelines do not recommend anti-TNFα drugs as chemopreventive agents.
**Ursodeoxycholic Acid**: The effect of UDCA is debated and controversial. In any case, it should not be used, especially at high doses, as a chemopreventive agent in patients affected by UC and PSC.
**Dietary compounds and lifestyle habits**: Even if a clear chemopreventive role of a specific diet or lifestyle habit has not been identified yet, some lifestyle strategies already validated for sporadic CRC, such as avoiding smoking and alcohol use and reducing red meat consumption, should be suggested.
**Statins**: Further studies are needed to confirm the potential role of statins in chemoprevention of IBS-associated CRC.
**Vitamin D**: Initial studies suggest a chemopreventive role for Vitamin D, but evidence is scarce. Given its high tolerability profile, it should be further investigated.
**Gut microbiome composition**: Since many alterations in the gut microbiome are involved in IBD pathogenesis, probiotics and prebiotics could have a potential role in the treatment of patients with IBD. Specific studies on their potential role in CRC prevention are needed.
**Secondary prevention**
**Open surveillance issues**: Endoscopic surveillance is an important prevention strategy; nevertheless, its effectiveness still needs to be demonstrated by RCTs.
**Timing of surveillance**: Surveillance colonoscopies should start 8 years after the onset of symptoms, at the time of diagnosis when PSC is present.
**Optimal endoscopic technique**: Enhanced dysplasia detection techniques (VCE or DCE) with non-targeted biopsies of non-suspicious areas and targeted biopsies of abnormalities should be performed.
**Management of dysplasia detection**: Grade of confirmed dysplasia (LGD vs. HGD) as well as its visibility and resectability are crucial. Colectomy is necessary in case of unresectable visible dysplasia or HGD or invisible multifocal dysplasia, while endoscopic polypectomy should be chosen if the lesions can be resected.
**Tertiary prevention**
CRC recurrence in patients with IBDs is rare. Surveillance could be proposed for patients with concomitant PSC or chronic pouchitis.

CRC = colorectal carcinoma; UC = ulcerative colitis; CD = Crohn’s disease; IBDs = inflammatory bowel diseases; TNFα = tumor necrosis Factor α; RCTs = randomized controlled trials; VCE = virtual chromoendoscopy; DCE = dye chromoendoscopy; HGD = high-grade dysplasia; LGD = low-grade dysplasia; PSC = primary sclerosing cholangitis.

## Data Availability

Data are contained within the article.

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
