# Peer review of "Colorectal Cancer in Inflammatory Bowel Diseases: Epidemiology and Prevention: A Review"

_cancers, 2022, doi:10.3390/cancers14174254_

Round 1

Reviewer 1 Report

TITLE

OK.

ABSTRACT

OK.

INTRODUCTION

OK.

MATERIAL AND METHODS

The authors state that “All titles and abstracts obtained by database search were screened by three independent authors (SK, SB and FL)...”. And then they point out that “Disagreements were solved by discussion with a third author (EM)”. Do you mean a fourth author?

EPIDEMIOLOGY

OK.

RISK FACTORS AND PRIMARY PREVENTION

Table 1. Summary of risk factor...; please state risks factors (in plural).

Table 1. Please explain ALL abbreviations in the footnote, including OR (odds ratio) and HR (hazard ratio).

Regarding duration of disease, the authors describe several studies suggesting that it greatly influences the risk of CRC, but a summary (regarding the appropriate cut-off point: 8-10 years?) for a clinical sound recommendation for clinical practice would be needed here.

Table 2. Please confirm that the information from the different columns of the table is properly aligned.

In the Thiopurines section, a relevant study is lacking: Gordillo J, Cabré E, Garcia-Planella E, Ricart E, Ber-Nieto Y, Márquez L, Rodríguez-Moranta F, Ponferrada Á, Vera I, Gisbert JP, Barrio J, Esteve M, Merino O, Muñoz F, Domènech E; ENEIDA Project of the Spanish Working Group in Crohn’s Disease and Ulcerative Colitis (GETECCU). Thiopurine Therapy Reduces the Incidence of Colorectal Neoplasia in Patients with Ulcerative Colitis. Data from the ENEIDA Registry. J Crohns Colitis. 2015 Dec;9(12):1063-70. doi: 10.1093/ecco-jcc/jjv145. Epub 2015 Sep 7. PMID: 26351379.

SECONDARY PREVENTION

It is stated that “This led to underlining the importance that the initiation of surveillance must take place from the moment of the appearance of symptoms and not from the date of diagnosis...” Please clarify that the authors are referring to the calculation of the timing for surveillance, and not that the surveillance should start right at the time symptoms appear.

European Crohn and Colitis Organization (please state Organisation, with s).

TERTIARY PREVENTION

OK.

CONCLUSIONS

OK.

REFERENCES

OK (see above the missing reference).

TABLES

OK (see above corresponding comments).

Author Response

Reviewer: 1

TITLE: OK.

ABSTRACT: OK.

INTRODUCTION:OK.

MATERIAL AND METHODS

The authors state that “All titles and abstracts obtained by database search were screened by three independent authors (SK, SB and FL)...”. And then they point out that “Disagreements were solved by discussion with a third author (EM)”. Do you mean a fourth author?

Reply: We are sorry for the oversight on our side, and we corrected the sentence.

EPIDEMIOLOGY: OK.

RISK FACTORS AND PRIMARY PREVENTION

Table 1. Summary of risk factor...; please state risks factors (in plural).

Table 1. Please explain ALL abbreviations in the footnote, including OR (odds ratio) and HR (hazard ratio).

Table 2. Please confirm that the information from the different columns of the table is properly aligned.

Reply: We thank the Reviewer for this suggestion. Tables were modified and Reviewer’s suggestions have been followed.

Regarding duration of disease, the authors describe several studies suggesting that it greatly influences the risk of CRC, but a summary (regarding the appropriate cut-off point: 8-10 years?) for a clinical sound recommendation for clinical practice would be needed here.

Reply: This aspect is extensively treated in the “Secondary prevention” section. Nevertheless, a sentence regarding the appropriate cut-off of 10 years has been inserted.

In the Thiopurines section, a relevant study is lacking: Gordillo J, Cabré E, Garcia-Planella E, Ricart E, Ber-Nieto Y, Márquez L, Rodríguez-Moranta F, Ponferrada Á, Vera I, Gisbert JP, Barrio J, Esteve M, Merino O, Muñoz F, Domènech E; ENEIDA Project of the Spanish Working Group in Crohn’s Disease and Ulcerative Colitis (GETECCU). Thiopurine Therapy Reduces the Incidence of Colorectal Neoplasia in Patients with Ulcerative Colitis. Data from the ENEIDA Registry. J Crohns Colitis. 2015 Dec;9(12):1063-70. doi: 10.1093/ecco-jcc/jjv145. Epub 2015 Sep 7. PMID: 26351379.

Reply: We would like to thank the Reviewer for pointing out this manuscript, that has now been included in the reference list and acknowledged in the revised version of the manuscript.

SECONDARY PREVENTION

It is stated that “This led to underlining the importance that the initiation of surveillance must take place from the moment of the appearance of symptoms and not from the date of diagnosis...” Please clarify that the authors are referring to the calculation of the timing for surveillance, and not that the surveillance should start right at the time symptoms appear.

Reply: We thank the Reviewer for pointing out this aspect, the sentence has been clarified.

European Crohn and Colitis Organization (please state Organisation, with s).

Reply: We thank the Reviewer for pointing out this misspelling, a correction has been made.

TERTIARY PREVENTION: OK.

CONCLUSIONS: OK.

REFERENCES: OK (see above the missing reference).

TABLES: OK (see above corresponding comments).

Reply: We would like to thank the Reviewer for her/his positive comments on our manuscript, and we do hope that the modifications made to the reviewed version will further improve our contribution

Reviewer 2 Report

Overall: editing for grammar/English needs to be completed.

Introduction/methods

-        Because this is a review, significantly more material is needed in the introduction and a methods section isn’t needed, unless this is being submitted as a systematic review and/or meta-analysis.

-       Please provide a summary of IBDs, e.g., etiology, characteristics, etc., to lead into CRC epidemiology

-      Search strategy terms can be put in a supplemental table

-      Is there a study flow chart? If this is a systematic review, that should be included.

Epidemiology

-        This section needs to be broken down into subsections. In its current form, there is a lot of information that doesn’t necessarily all work together. Perhaps it would be better to have a subsection for IBD/CD, then one for IBD-associated CRC.

Risk factors and primary prevention

-        Table 1 – since the authors went to the trouble of performing a systematic review, it may be helpful to calculate summary estimates of risk factors rather than including an already published table. The current table doesn’t add any new information.

-        Table 2 – This table is not particularly helpful, it is just a list of studies. Please provide the primary studies, or, at a minimum, the summary estimates, so the reader is able to see the results of the previous studies. Further, reporting primary studies, rather than meta-analyses, would be more helpful.

-        Why are lifestyle/behavioral interventions not included under primary prevention?

Secondary prevention

-        This section needs to be edited to be more concise. There is too much information that it is not easy for the reader to understand the point of the section.

-        It may be important to note that cancer prevention agents have also been evaluated for their efficacy for secondary and tertiary prevention. At least a summary of these studies should be included, and how they may be helpful for prevention of CRC in IBD/CD.

Conclusions

-        The conclusions are severely lacking. This would be a great spot to get into the limitations of the current methods and propose directions the field could go.

Author Response

Reviewer 2:

Overall: editing for grammar/English needs to be completed.

Reply: We thank the Reviewer for this suggestion. Grammar and English revision has been made and mistakes have been corrected.

Introduction/methods

-        Because this is a review, significantly more material is needed in the introduction and a methods section isn’t needed, unless this is being submitted as a systematic review and/or meta-analysis.

Reply: We thank the Reviewer for this suggestion. Introduction section has been developed as suggested and the adequate references have been acknowledged and added to the reference list. Methods section has been modified as suggested by Reviewer 3.

-       Please provide a summary of IBDs, e.g., etiology, characteristics, etc., to lead into CRC epidemiology.

Reply: We thank again the Reviewer for this suggestion.  A synthetic summary of IBDs has been introduced in the section.

-      Search strategy terms can be put in a supplemental table

Reply: We do fully agree with this comment and following the Reviewer’s suggestion we have created Supplementary Table S1 which contains all the strategy terms divided per Database.

-      Is there a study flow chart? If this is a systematic review, that should be included.

Reply: This is a NonSystematic Review, also known as Narrative Review. For this reason, the Study Flow Chart was not included.

Epidemiology

-        This section needs to be broken down into subsections. In its current form, there is a lot of information that doesn’t necessarily all work together. Perhaps it would be better to have a subsection for IBD/CD, then one for IBD-associated CRC.

Reply: We thank the Reviewer for this suggestion. Epidemiology section has been divided into subsections as suggested.

Risk factors and primary prevention

-        Table 1 – since the authors went to the trouble of performing a systematic review, it may be helpful to calculate summary estimates of risk factors rather than including an already published table. The current table doesn’t add any new information.

Reply: As stated in a previous Reply, this is a NonSystematic Review, also known as Narrative Review.

-        Table 2 – This table is not particularly helpful, it is just a list of studies. Please provide the primary studies, or, at a minimum, the summary estimates, so the reader is able to see the results of the previous studies. Further, reporting primary studies, rather than meta-analyses, would be more helpful.

Reply: We do fully agree with this comment and following the Reviewer’s suggestion we have modified the Table. After doing that, we realized that reporting all the primary studies would have made the Table too long especially since, as it is now clarified in the Introduction, our aim is to give clear key messages to the reader. Therefore, we have opted for inserting summary estimates as suggested by the Reviewer,

-        Why are lifestyle/behavioral interventions not included under primary prevention?

Reply: Paragraph “4.2.6. Dietary compounds and lifestyle habits” is part of Section “4. Risk Factors and Primary Prevention”

Secondary prevention

-        This section needs to be edited to be more concise. There is too much information that it is not easy for the reader to understand the point of the section.

Reply: We thank the Reviewer for this suggestion. A sentence reporting the Key message of the section has been added to provide a clear summary of the paragraph to help the reader to understand the point of the section.

-        It may be important to note that cancer prevention agents have also been evaluated for their efficacy for secondary and tertiary prevention. At least a summary of these studies should be included, and how they may be helpful for prevention of CRC in IBD/CD.

Reply: We agree with the Reviewer’s comment, and therefore the section “Tertiary Prevention” has been modified as suggested.

Conclusions

-        The conclusions are severely lacking. This would be a great spot to get into the limitations of the current methods and propose directions the field could go.

Reply: We agree with the Reviewer’s comment, and therefore the section “Conclusion” has been enriched as suggested. Furthermore, following Reviewer’s hint we have modified the title of the last paragraph from “Conclusions” to “Conclusions and future perspectives”.

Reviewer 3 Report

Authors tried to make a Review regarding Epidemiology and Prevention of Colorectal Cancer in Patients with Inflammatory Bowel Diseases. The topic is much too generous to be comprised in 10 pages of MDPI style, where a page is 2/3 occupied with text. Extensive revision is needed and work. Please see below my suggestions, in order to improve this manuscript:

Introduction is much too poor. I suggest few ideas that can enrich this part.

First, more data must be added regarding the colon cancer, as follows:

-What importance heave early diagnosis of colon cancer?

- How can it be diagnosed earlier (techniques, apparatus, approaches)? 

- Which treatments are provided in protocols (i.e. immunohistochemical and histoenzymatic techniques, etc.) and what modern therapies are suggested by the literature (plant based, nanotechnologies) for colon cancer? A separate paragraph must be dedicated to colon cancer, diagnostics and treatment options. In this regard, I suggest checking and referring to: https://pubmed.ncbi.nlm.nih.gov/26662146/ ; https://doi.org/10.3390/molecules27072129 ; https://doi.org/10.3390/biom11081176 ; https://doi.org/10.1007/s11356-020-09028-0

Second, a solid paragraph must be added regarding the Bowel Diseases, in the same way as above (diagnosis, treatments) – I suggest DOI: 10.1016/j.jfma.2018.07.005   ; DOI: 10.1093/ecco-jcc/jjy114 ; etc.

L49-51. Aim of the study is very poor and must be better developed. As the topic is not a new one, it is needed of highlighting some aspects by responding to the following questions: Which is the novelty of your study or the special aspects it brings to the field? What makes different your study from others in the same/similar topic, already published? Why have the authors have chosen this topic as the literature is plenty of details on this topic?

2. Materials and Methods. As this is a Review, the titles of the sections must be chosen as to best fit their content. In this case, I suggest replacing the title of the section with Literature data selection, or something similar), 

Almost the entire text in this section must be included in a PRISMA graph. As the authors have stated that this is a Review, a PRISMA flow chart is recommended. I suggest checking both Page et al. papers, where this type of graphic is very well described: Page, M.J.; McKenzie, J.E.; Bossuyt, P.M.; Boutron, I.; Hoffmann, T.C.; Mulrow, C.D.; Shamseer, L.; Tetzlaff, J.M.; Akl, E.A.; Brennan, S.E.; et al. The PRISMA 2020 statement: An updated guideline for reporting systematic reviews. Journal of Clinical Epidemiology 2021, 134, 178-189, doi:10.1016/j.jclinepi.2021.03.001. Page, M.J.; McKenzie, J.E.; Bossuyt, P.M.; Boutron, I.; Hoffmann, T.C.; Mulrow, C.D.; Shamseer, L.; Tetzlaff, J.M.; Moher, D. Updating guidance for reporting systematic reviews: development of the PRISMA 2020 statement. Journal of Clinical Epidemiology 2021, 134, 103-112, doi:10.1016/j.jclinepi.2021.02.003.  and will help you to provide a correct PRISMA flow chart. Please take care and detail in the best way the inclusion/exclusion criteria used for the literature selection. Include here also the MeSH terms. Do not forget renumbering the following sections.

Remove the last sentence of the section. Nobody is interested how were solved the disagreements between the authors.

Table 1: last column of the table must be inserted Ref. (references)

Table 2: Remove the 3rd column and insert as above, last column Ref.

Sub-subsection 4.2.4. What can be the relevance of a subsection having 1 line (L268)? Please delete this subsection. Moreover, all the 4th’ subsections must be better developed. For 4.2.7. I suggest checking and developing more ideas the authors can find in  https://doi.org/10.3390/diagnostics11061090

The part with colonoscopy screening strategies needs to be better discussed and summarized with a table where you include results of published met-analysis.

Table 3. Complete properly the head of the table for the 3rd and 4th column. No need to bold text in the table’s rows, only the title must be bolded.

Please develop a new subsection where to detail the role of other important substances in the prevention of colorectal cancer. Discuss the role of vitamin D supplementation in colorectal cancer prevention. Which is the role of metformin, and of the statins in colorectal cancer prevention?

Conclusions part contains information provided in the sections, in duplicate. Please reconsider, providing only authors opinions about this topic.

Author Response

Authors tried to make a Review regarding Epidemiology and Prevention of Colorectal Cancer in Patients with Inflammatory Bowel Diseases. The topic is much too generous to be comprised in 10 pages of MDPI style, where a page is 2/3 occupied with text. Extensive revision is needed and work. Please see below my suggestions, in order to improve this manuscript:

Introduction is much too poor. I suggest few ideas that can enrich this part.
First, more data must be added regarding the colon cancer, as follows:

-What importance heave early diagnosis of colon cancer?

- How can it be diagnosed earlier (techniques, apparatus, approaches)? 

- Which treatments are provided in protocols (i.e. immunohistochemical and histoenzymatic techniques, etc.) and what modern therapies are suggested by the literature (plant based, nanotechnologies) for colon cancer? A separate paragraph must be dedicated to colon cancer, diagnostics and treatment options. In this regard, I suggest checking and referring to: https://pubmed.ncbi.nlm.nih.gov/26662146/ ; https://doi.org/10.3390/molecules27072129 ; https://doi.org/10.3390/biom11081176 ; https://doi.org/10.1007/s11356-020-09028-0

Reply: We thank the Reviewer for his suggestion to enrich the Introduction. It has been modified as suggested and now contains an introductory summary of IBDs, CRC and CRC in IBDs. Moreover, in the Introduction we have clarified our intention to focus on the epidemiology and prevention of CRC specifically in patients affected by IBDs. We do agree with the Reviewer’s feeling that aspects concerning diagnostics and treatment options of sporadic CRC are important. However, since our aim is to give clear key messages about the management of preventive strategies in patients with inflammatory bowel diseases, we decided to refer the reader to the numerous reviews on those topics already available in the literature.

Second, a solid paragraph must be added regarding the Bowel Diseases, in the same way as above (diagnosis, treatments) – I suggest DOI: 10.1016/j.jfma.2018.07.005   ; DOI: 10.1093/ecco-jcc/jjy114 ; etc.

Reply: As per our previous reply, we thank the Reviewer for his comment and a specific paragraph introducing IBDs has been added. At the same time, we think that IBDs diagnosis and treatment are topics much too generous to be comprised in the introduction of a Review dealing specifically with a heterogeneous theme such as epidemiology and Prevention of CRC in patients with IBDs.

L49-51. Aim of the study is very poor and must be better developed. As the topic is not a new one, it is needed of highlighting some aspects by responding to the following questions: Which is the novelty of your study or the special aspects it brings to the field? What makes different your study from others in the same/similar topic, already published? Why have the authors have chosen this topic as the literature is plenty of details on this topic?

Reply: We do fully agree with this comment and following the Reviewer’s suggestion we have clarified the aims of our work. In particular, it is now explicated our will to simplify the misleading and heterogeneous literature’s evidences in order to deliver to the reader an updated key of interpretation.

  1. Materials and Methods. As this is a Review, the titles of the sections must be chosen as to best fit their content. In this case, I suggest replacing the title of the section with Literature data selection, or something similar), 

Reply: We do fully agree with this comment and following the Reviewer’s suggestion we have modified the title of the section as suggested.

Almost the entire text in this section must be included in a PRISMA graph. As the authors have stated that this is a Review, a PRISMA flow chart is recommended. I suggest checking both Page et al. papers, where this type of graphic is very well described: Page, M.J.; McKenzie, J.E.; Bossuyt, P.M.; Boutron, I.; Hoffmann, T.C.; Mulrow, C.D.; Shamseer, L.; Tetzlaff, J.M.; Akl, E.A.; Brennan, S.E.; et al. The PRISMA 2020 statement: An updated guideline for reporting systematic reviews. Journal of Clinical Epidemiology 2021, 134, 178-189, doi:10.1016/j.jclinepi.2021.03.001. Page, M.J.; McKenzie, J.E.; Bossuyt, P.M.; Boutron, I.; Hoffmann, T.C.; Mulrow, C.D.; Shamseer, L.; Tetzlaff, J.M.; Moher, D. Updating guidance for reporting systematic reviews: development of the PRISMA 2020 statement. Journal of Clinical Epidemiology 2021, 134, 103-112, doi:10.1016/j.jclinepi.2021.02.003.  and will help you to provide a correct PRISMA flow chart. Please take care and detail in the best way the inclusion/exclusion criteria used for the literature selection. Include here also the MeSH terms. Do not forget renumbering the following sections.

Reply: This is a NonSystematic Review, also known as Narrative Review. For this reason PRISMA guidelines, specific for systematic reviews, were not followed.

Remove the last sentence of the section. Nobody is interested how were solved the disagreements between the authors.

Reply: We have followed the suggestion of the Reviewer.

Table 1: last column of the table must be inserted Ref. (references)

Table 2: Remove the 3rd column and insert as above, last column Ref.

 Reply: Following the Reviewer’s comments we have deeply modified both the tables.

Sub-subsection 4.2.4. What can be the relevance of a subsection having 1 line (L268)? Please delete this subsection. Moreover, all the 4th’ subsections must be better developed.

Reply: We thank the reviewer for pointing out these aspects. The mentioned section has been removed, as suggested. All the 4 subsections have been developed and deeply improved, as suggested.

For 4.2.7. I suggest checking and developing more ideas the authors can find in  https://doi.org/10.3390/diagnostics11061090

Reply: We would like to thank the Reviewer for pointing out this manuscript, that has now been included in the reference list and acknowledged in the revised version of the manuscript

The part with colonoscopy screening strategies needs to be better discussed and summarized with a table where you include results of published met-analysis.

Reply: We thank the Reviewer for his/her suggestions. Table 3 is based on evidence-based recommendations made by the most influent gastroenterological scientific societies of the world. In our opinion, delivering to the reader the practical recommendations by those societies is more helpful In order to give him an interpretation key of several meta-analysis. At the same time, we share the reviewer opinion about implement clear messages on endoscopic surveillance in Tables. Also for this purpose, we have created a Key messages’ Table, containing also all the core-tips about colonoscopy screening strategies.

 Table 3. Complete properly the head of the table for the 3rd and 4th column. No need to bold text in the table’s rows, only the title must be bolded.

Reply: We have followed the suggestion of the Reviewer.

Please develop a new subsection where to detail the role of other important substances in the prevention of colorectal cancer. Discuss the role of vitamin D supplementation in colorectal cancer prevention. Which is the role of metformin, and of the statins in colorectal cancer prevention?

Reply: We do fully agree with this comment and following the Reviewer’s suggestion we have introduced the analysis of chemopreventive role of statins and vitamin D.

Conclusions part contains information provided in the sections, in duplicate. Please reconsider, providing only authors opinions about this topic.

Reply: We agree with the Reviewer’s comment, and therefore the section “Conclusion” has been enriched as suggested. Furthermore, following Reviewer’s hint we have modified the title of the last paragraph from “Conclusions” to “Conclusions and future perspectives”.

Round 2

Reviewer 2 Report

Marabotto et al. present a revised version of a review evaluating the epidemiology of IBD-associated CRC. Though some portions of the review are improved, significant work is required to make this review suitable for publication. Major comments are listed below.

Again, please heavily edit for grammar, word choice, and sentence structure. There remain too many errors to list here. Furthermore, the sections/subsections should be more concise and summarized. In it’s current form, the review is primarily a list of studies and the individual results, making it difficult to discern the point. Pulling out individual examples, especially for seminal studies, is fine, but the entire article cannot be individual studies. Please clean this up.

L15 – Change ‘on the lights of’ to ‘in light of’, which I believe is what is intended.

Introduction

The introduction is disorganized and contains a lot of information for a very short section. Please edit to be more concise.

Literature Research

L73 – A more appropriate title for this section would be ‘Literature Review’

I’m still confused why outlining the search strategy is needed. As the authors have stated in their reply that this is a narrative review, I don’t understand why it was structured like a systematic. Please address this and perhaps edit the methods.

The inclusion of ‘Key Messages’ at the end of each subsection is unnecessary and redundant. Table 4 is fine but needs to be edited.

L131-136 – Costs of IBD don’t belong in a subsection with epidemiology. If you are talking about costs of treatment or financial toxicity, that is its own discussion.

L143-159 – Summarize the IRs. There is no need to individually address studies.

L160-162 – Rewrite and include in the previous paragraph.

L201-203 – Again, please take out the ‘Key Message’. If you decide to leave these in, the key message of subsection 3.3 does not match the content. There is no discussion of surveillance or endoscopy, so you cannot conclude that is why incidence is decreasing.

L207-209 – Reorganize sentence.

Table 1 – It is misleading to include individual study estimates and meta-analyses in a summary table. Unless you are calculating a summary OR/HR (as in a meta-analysis), this could be cherry-picking studies. I would rethink the format and message of this table. Further, the ‘Burden’ column is redundant. The same information is shown in the OR and HR columns.

Table 2 – Include the summary estimates from the meta-analyses in the table.

L264 – Change ‘a review from Nguyen et al. published in 2012’ to something like ‘a conflicting review’. The extra detail is not needed.

L414-431 – This subsection, in particular, is very vague. Either include more concrete examples of how gut alterations may impact IBD-associated CRC or delete.

Author Response

Marabotto et al. present a revised version of a review evaluating the epidemiology of IBD-associated CRC. Though some portions of the review are improved, significant work is required to make this review suitable for publication. Major comments are listed below.

Again, please heavily edit for grammar, word choice, and sentence structure. There remain too many errors to list here. Furthermore, the sections/subsections should be more concise and summarized. In it’s current form, the review is primarily a list of studies and the individual results, making it difficult to discern the point. Pulling out individual examples, especially for seminal studies, is fine, but the entire article cannot be individual studies. Please clean this up.

Reply: In this updated version of the manuscript all the above recommendations of Reviewer 2 have been followed. In particular, a deep revision for grammar, word choice and sentence structure has been carried out. Furthermore, all the sections have been revised and are now more concise and less “wordy”. More focus was put on the message and the rationale of the studies cited rather than presenting them as a list.

We do hope that the modifications made to the reviewed version will further improve our contribution.

L15 – Change ‘on the lights of’ to ‘in light of’, which I believe is what is intended.

Reply: We thank the reviewer for his/her suggestion, the correction has been made.

Introduction

The introduction is disorganized and contains a lot of information for a very short section. Please edit to be more concise.

Reply: The introduction section has been modified according to Reviewers’ suggestions (see comments of Reviewer 1 and Reviewer 3) who asked for more information regarding IBD, colon cancer and their association. Thus, we have implemented it with “significantly more data material” providing a “summary of IBDs, e.g., etiology, characteristics, etc” as specifically suggested in the first round of Reviewer 1 comments (but also Reviewer 3). Since both Reviewers 1 and 3 agreed on changes made, we feel it may be inadequate to further modify this section. Nevertheless, according to this Reviewer’s suggestion, we tried to be more concise.

Literature Research

L73 – A more appropriate title for this section would be ‘Literature Review’

I’m still confused why outlining the search strategy is needed. As the authors have stated in their reply that this is a narrative review, I don’t understand why it was structured like a systematic. Please address this and perhaps edit the methods.

Reply: The Title of the Section is now “Literature Review”, as suggested. As described in one of the most cited manuscripts in the literature dealing with how a narrative review should be written (DOI: 10.1016/S0899-3467(07)60142-6), we feel that this section should describe “step by step” how the study was performed. In any case, modifications have been made following Reviewer 2 and 3 and this section is now more concise.

The inclusion of ‘Key Messages’ at the end of each subsection is unnecessary and redundant. Table 4 is fine but needs to be edited.

Reply: Following the Reviewer indication, Key Messages have been deleted at the end of each paragraph and Table 4 has been edited as suggested.

L131-136 – Costs of IBD don’t belong in a subsection with epidemiology. If you are talking about costs of treatment or financial toxicity, that is its own discussion.

Reply: We agree with the Reviewer’s comment, and therefore the costs of IBDs have been deleted from this section.

L143-159 – Summarize the IRs. There is no need to individually address studies.

L160-162 – Rewrite and include in the previous paragraph.

Reply: We agree with the Reviewer’s comments. IRs and studies have been summarized as suggested. The L160-162 have been included in the previous paragraph as suggested.

L201-203 – Again, please take out the ‘Key Message’. If you decide to leave these in, the key message of subsection 3.3 does not match the content. There is no discussion of surveillance or endoscopy, so you cannot conclude that is why incidence is decreasing.

Reply: As previously reported, Key Messages have been deleted at the end of each paragraph.

L207-209 – Reorganize sentence.

Reply: We thank the Reviewer for his suggestion. The sentence has been reorganized.

Table 1 – It is misleading to include individual study estimates and meta-analyses in a summary table. Unless you are calculating a summary OR/HR (as in a meta-analysis), this could be cherry-picking studies. I would rethink the format and message of this table. Further, the ‘Burden’ column is redundant. The same information is shown in the OR and HR columns.

Reply: We agree with the Reviewer’s comment. Table1 has been deleted following Reviewer’s suggestion, all the studies and their rationale are now clearly described in the text.

Table 2 – Include the summary estimates from the meta-analyses in the table.

Reply: We do fully agree with this comment and following the Reviewer’s suggestion we have modified the Table.

L264 – Change ‘a review from Nguyen et al. published in 2012’ to something like ‘a conflicting review’. The extra detail is not needed.

Reply: The sentence has been modified as suggested by the Reviewer

L414-431 – This subsection, in particular, is very vague. Either include more concrete examples of how gut alterations may impact IBD-associated CRC or delete.

Reply: We agree with the Reviewer comments. This section has been updated with more concrete examples as suggested. It has to be emphasized that real-life evidence on this issue is scanty, and most of the published studies are mainly focused on hypotheses or experimental studies. Since this review was aimed to provide clear key messages about the management of preventive strategies in patients with IBDs, we feel that a complete dissertation about the hypothetical role of microbiota alterations in IBD-associated CRC is a topic much too generous to be fully elucidated in our paper.

Reviewer 3 Report

The authors made some corrections, but the manuscript is still not well developed/structured. Some points I mentioned in my previous report (please check it again and proceed) have been not properly addressed. Please pay more attention to the following aspects:

L71-72. Aim of the study remained poor. No novelty was highlighted.

I suggest inserting The detailed web-research in the manuscript, in the 2nd section, not as supplementary material.

L98-99. No needing mentioning "All titles and abstracts obtained by database search were screened by four independent authors (SK, SB and, FL and EM) and". Please remove. Authors' contribution is a special section at the final of the manuscript.

No needing highlighting the last sentence of each sections 3 to 7 as key message. Remove "Key message" terms at the beginning of each such statement.

L602-607 should be inserted in one of the section above (maybe in a new one named 8. Discussion), but not in Conclusions. never Table in Conclusions. Conclusion must be a single paragraph (maximum 2), briefly describing the authors' opinions regarding their own conclusions after performing a research. Future directions can be also mentioned in the Conclusions. Moreover, NO references should be mentioned in this part, for the same reason that Conclusions are of the authors.

L634. remove "we believe". It is not a scientific approach to believe, but to explain or state real things.

References section must be updated, as some refs are older than 40 years! (1980?) Check the recent references I suggest in my previous report (which are not mine); you will find them very useful in enriching this reference section.

Author Response

The authors made some corrections, but the manuscript is still not well developed/structured. Some points I mentioned in my previous report (please check it again and proceed) have been not properly addressed.

Please pay more attention to the following aspects:

L71-72. Aim of the study remained poor. No novelty was highlighted.

Reply: Following the Reviewer suggestions, the aim of the study has been enriched and innovations, especially regarding effects of new therapies and endoscopic techniques, clarified.

I suggest inserting The detailed web-research in the manuscript, in the 2nd section, not as supplementary material.

Reply: As already known the detailed web-research was originally reported in this section. However, we received contrasting suggestions by the Reviewers about it. Since other Reviewers suggested to delete completely the search methods, we think that including it in Supplementary Material could be a fair compromise.

L98-99. No needing mentioning "All titles and abstracts obtained by database search were screened by four independent authors (SK, SB and, FL and EM) and". Please remove. Authors' contribution is a special section at the final of the manuscript.

Reply: We thank the Reviewer for pointing out this aspect. The sentence has been removed as suggested.

No needing highlighting the last sentence of each sections 3 to 7 as key message. Remove "Key message" terms at the beginning of each such statement.

Reply: Key Messages have been deleted at the end of each paragraph.

L602-607 should be inserted in one of the section above (maybe in a new one named 8. Discussion), but not in Conclusions. never Table in Conclusions. Conclusion must be a single paragraph (maximum 2), briefly describing the authors' opinions regarding their own conclusions after performing a research. Future directions can be also mentioned in the Conclusions. Moreover, NO references should be mentioned in this part, for the same reason that Conclusions are of the authors.

Reply: We thank the Reviewer for his suggestions, the paragraphs has been modified and all the corrections suggested by the Reviewer were followed.

L634. remove "we believe". It is not a scientific approach to believe, but to explain or state real things.

References section must be updated, as some refs are older than 40 years! (1980?) Check the recent references I suggest in my previous report (which are not mine); you will find them very useful in enriching this reference section.

Reply L634 has been modified as suggested. As far as dated references are concerned, the vast majority of the references were published in the last ten years, and only one was published before 1980: the study by Dr. Crohn himself published in 1925. Regarding the references suggested by the Reviewer in the previous round of comments they have been carefully evaluated and taken into account, and https://doi.org/10.3390/diagnostics11061090 has already been included in the manuscript after the first round of comments. Furthermore, following Reviewer 3 suggestions, 10.1093/ecco-jcc/jjy114 and 10.1016/j.jfma.2018.07.005 have been also included in the manuscript. Thanks for this additional suggestions.

Round 3

Reviewer 2 Report

The edits provided by Marabotto et al. are sufficient. I would still recommend thoroughly editing the manuscript prior to publication, as there remains significant grammatical issues.

Author Response

We thank the Reviewer for his/her constructive criticism that helped us in improving our work. In this updated version a revision for grammar, word choice and sentence structure has been carried out as suggested.

We do hope that the modifications made to the reviewed version will further improve our contribution.

Reviewer 3 Report

The authors responded to my requests.

Author Response

We thank the Reviewer for his/her constructive criticism that helped us in improving our work.